# Does the Integrated Development of Agriculture and Tourism Promote Farmers' Income Growth? Evidence from Southwestern China

**Yuxi Luo [1,2,3], Tianren Xiong [1], Defeng Meng [1,2], Anrong Gao [1,2,\*] and Yan Chen [4,\*]**

[1] School of Economics and Management, Guangxi Normal University, Guilin 541006, China; yluogxnu@mailbox.gxnu.edu.cn (Y.L.); xiongtr@stu.gxnu.edu.cn (T.X.); mdf_gxnu@mailbox.gxnu.edu.cn (D.M.)

[2] Pearl River-Xijiang River Economic Belt Development Institute, Guangxi Normal University, Guilin 541004, China

[3] Guangxi Key Laboratory of Landscape Resources Conservation and Sustainable Utilization in Lijiang River Basin, Guangxi Normal University, Guilin 541004, China

[4] School of Business, Guilin Tourism University, Guilin 541006, China

\* Correspondence: gaoanrong@mailbox.gxnu.edu.cn (A.G.); cylove626@163.com (Y.C.)

**Abstract:** The integrated development of agriculture and tourism is an effective driving force to boost farmers' income. We utilize a quasi-natural experiment design to test how such integrated development promotes the comprehensive rural revitalization. By adopting a panel dataset of 72 counties within Guangxi province from 2005 to 2020 and a PSM-DID method, we attempt to explore the effect of the integrated development of agriculture and tourism on farmers' income growth. The empirical results support our hypothesis that the integrated development of agriculture and tourism can effectively promote farmers' income growth and its regional heterogeneity with respect to tourism resource endowment and economic development level. We further discuss the transmission mechanism of the integrated development of agriculture and tourism and reveal that the agricultural technology level and agricultural production efficiency have mediating effects on improving farmers' income growth. However, a masking effect exists between the integrated development of agriculture and tourism and the level of non-agricultural employment. The possible reason is that industrial and commercial capital investment has crowded out the welfare originally belonging to the wage income and only allowed farmers to obtain the one-time land rent income.

**Keywords:** integrated development; farmers' income growth; PSM-DID; masking effect

## 1. Introduction

The integration of agriculture and tourism can date back to Germany's "Civic Paradise" in the 1850s. Today, the integrated development of agriculture and tourism refers to a combination of agriculture and tourism industries to form a new setting of comprehensive economic activities (Streimikiene, 2015) [1]. This integrated approach unites agricultural and tourism resources through various means like agricultural sightseeing, rural tourism, leisure agriculture, homestays, etc. This synthesis promotes resource sharing, enhances resource utilization efficiency, and stimulates the coordinated growth of related sectors, ultimately yielding economic, social, and environmental benefits for rural areas and driving farmers' income growth (Barbieri, 2013) [2]. As an important factor to promote the development of rural areas and boost regional rural economy, the integrated development of agriculture and tourism has become a fundamental measure to deepen the industrial integration in rural areas, driving more attention by the government. With respect to the regional differences, and the integration of agriculture and tourism has formed heterogeneity in industrial modes and development concepts. The integration of agricultural resources and tourism management concepts helps to enhance the market awareness and

competitiveness of agricultural products, thereby improving the added value of agricultural resources. Consequently, it always been considered as an important means to promote farmers' income (Xiao, 2017) [3]. This has led the academic community to undertake extensive research on related topics. The initial research mainly focused on the conceptual connotation of the integration of agriculture and tourism (Briedenhann and Wickens, 2004; Zhang and Chen, 2009; Phillip and Hunter, 2010; Arroyo et al., 2013) [4–7], agriculture and tourism integration modes (Zheng et al., 2022; Li and Wang, 2022; Jiang, 2021) [8–10], its development path and integration level measurement (Zhang, 2022; Ouyang and Li, 2018; Lin et al., 2022; Yang et al., 2020; Zhou et al., 2016), and other aspects [11–15].

The combination of agricultural resources and tourism management concepts can significantly enhance the added value of agricultural resources, rendering it a consistent means of augmenting farmers' income (Gu et al., 2021) [16]. Especially in recent years, with the continuous deepening of the implementation of China's rural revitalization strategy, rural infrastructure has undergone substantial enhancements, public services have grown more comprehensive, and the foundational conditions for integrating agriculture and tourism have steadily improved. As a means of raising farmers' incomes and promoting non-agriculture employment, local governments are increasingly supporting to the integrated development of agriculture and tourism (Tew and Barbieri, 2012; Suardana and Sudiarta, 2017) [17,18]. Although the integration of agriculture and tourism can improve the efficiency of resource allocation and thus increase the added value through industrial integration, there is no clear answer on whether this added value can be converted into farmers' non-agricultural income. Studies have indeed explored the potential for farmers' income growth through the integration of agriculture and tourism. However, the existing literature has yet to provide a comprehensive explanation for the specific pathways through which this income enhancement occurs.

Traditional agriculture faces constraints such as limited land resources, technology, and labor, leading to diminishing marginal returns that restrict the growth efficiency of agricultural income. In contrast, the integration of agriculture and tourism industry introduces a fresh avenue for growth (Zhang et al., 2023) [19]. In terms of augmenting farmers' income, the tourism industry, rather than non-agricultural products, offers greater potential for expansion. The improvement of non-agricultural employment often means the optimization and upgrading the structures of rural economic industries, and the improvement of non-agricultural production efficiency is an important factor to ensure the sustainable growth of farmers' income (Hu et al., 2022) [20]. However, it is worth noting that the participation of tourism industry alone does not mean the increase in local non-agricultural employment income level, primarily due to the constraints imposed by various social capital limitations in the region (Nugraha et al., 2022) [21]. Therefore, the significance of policy analysis on the mechanism of integration of agriculture and tourism on farmers' income growth based on non-agricultural employment is richer than that of traditional mechanism studies. The superior natural conditions and profound endowment of human resources in Guangxi provide high-quality cultivation soil and development conditions for the integrated development of agriculture and tourism, which is an excellent quasi-experimental sample to test how the integrated development of agriculture and tourism can promote the growth of farmers' income in southwestern China. On top of using traditional framework to confirm the promoting effect of the integrated development of agriculture and tourism on farmers' income growth, this paper conducts further research from a new perspective of employment structure by introducing non-agricultural employment variables under a PSM-DID setting, and it tries to answer the question of whether or not that the integrated development of agriculture and tourism really improves farmers' income growth with the promotion of the policy of agricultural and tourism integration in Guangxi Demonstration County from the new perspective of employment structure. Meanwhile, it tries to clarify the mechanism of the integrated development of agriculture and tourism in improving farmers' income, and it provides beneficial policy enlightenment for the comprehensive realization of a rural revitalization strategy.

## 2. Literature Review

The academic community has engaged in extensive discussions from various perspectives regarding the potential of agriculture and tourism integration to stimulate farmers' income growth. Differing opinions exist regarding the impact of this integration on farmers' income.

### 2.1. Implementation Effect of the Integrated Development of Agriculture and Tourism

Firstly, from the perspective of the effect of the integration of agriculture and tourism towards the regional economic development, Privitera (2009) suggests that the integration of agriculture and tourism can create employment and increase economic benefits [22]. It plays an important role in delaying the reduction in the rural population, stimulating employment, increasing non-agricultural farmers' income, narrowing the income gap between urban and rural areas, etc. (Hwang and Lee, 2015; Lupi et al., 2017; Hu and Wang, 2017; Liu et al., 2017) [23–26]. Moreover, it contributes to regional sustainability and environmentally friendly growth (Liu et al., 2023; Wang et al., 2022) [27,28]. Secondly, the research relating to its poverty alleviation effect has revealed that eliminating poverty by solely relying on agricultural production has natural shortcomings. Since agriculture and tourism are naturally complementary industries, multi-industry coordinated development plays an important role in poverty reduction (Shan et al., 2017) [29]. Because the integration of agriculture and tourism has realized the transformation from the single-industry to the multi-industry integrated development, it can increase consumers of agricultural products, create rural employment opportunities, and improve the living standards in low-income areas (Tew and Barbieri, 2012; Suardana and Sudiarta, 2017) [17,18]. Due to its strong penetration characteristics, tourism has the advantages of deep integration with other industries, and its multiplier effect also plays a positive role in poverty reduction (Mitchell and Phucl, 2007) [30]. Thirdly, in terms of the agricultural production efficiency, it is no doubt that agricultural innovation is a potential way to promote rural revitalization and reduce agricultural pollution (Liu et al., 2021) [31]. The integration of agriculture and tourism improves agricultural production efficiency through the accumulation of agricultural technology capital, and it plays a positive role in promoting sustainable agricultural development (Hu and Zhong, 2019; Xu et al., 2023; Zhuang et al., 2021) [32–34]. Although the integrated development mode of agriculture and tourism has injected vitality into rural supply-side reform, the integrated development of agriculture and tourism is characterized by heterogeneity due to different production technology levels and resource endowments (Tao, 2019) [35]. Fourthly, regarding to its impact on rural industrial structure optimization, the integration of agriculture and tourism plays a positive role in the transformation and upgrading of the primary and tertiary industries (Ning, 2014; Xu, 2013) [36,37]. Zhong et al. (2020) suggests that such integration can promote the optimization and upgrading of rural industrial structure by guiding consumption and increasing capital accumulation [38]. Furthermore, the integration of agriculture and tourism fosters social capital and community engagement, enabling communities to effectively respond to large-scale crises like epidemics and maintain its sustainable development (Prayitno et al., 2022) [39]. Although recent research perspectives primarily focus on the effects of the integration of agriculture and tourism on regional economic growth, agricultural labor efficiency, green poverty reduction and industrial structure optimization, there is a notable gap in discussing the promotion of this integration on macroeconomic development and micro labor production efficiency. In addition, few studies delve into the mechanism analysis and verification of the income growth effect resulting from the integrated development of agriculture and tourism.

### 2.2. Impact of Integration of Agriculture and Tourism on Farmers' Income

There is still controversy about whether the integrated development of agriculture and tourism can promote the farmers' income growth. From the perspective of mechanism, one view holds that the integration of agriculture and tourism can increase farmers' income

by creating more employment opportunities, improving agricultural labor production efficiency, promoting sales of local agricultural products, optimizing rural industrial structure, etc. (Li et al., 2018; Everett and Slocum, 2013; Kline et al., 2016; Cunha et al., 2020; Zhong and Tang, 2020) [40–44]; The other view is that the integrated development of agriculture and tourism will destroy the rural ecological environment and damage farmers' long-term interests. Moreover, the instability of the rural tourism market, limited rural tourism resources, and insufficient investment in the integrated development of agriculture and tourism will cause farmers' lack of confidence, thus affecting their participation in such practices. Consequently, the potential income-boosting effect of this integration may not fully materialize (Mastronardi et al., 2015; Islam and Carlsen, 2016; Barbieri, 2020; Hochuli et al., 2021) [45–48]. In terms of macro research, Kyu-Sok (2006) verified the income-increasing effect of the integration of agriculture and tourism on non-agricultural income by using secondary EU government data [49]. Xiao (2014) used China's provincial data to test the promoting effect of the integration of agriculture and tourism on the increase in farmers' income through the Spatial Durbin Model [50]. However, Manuel et al. (2015) believed that the integrated development of agriculture and tourism would bring about negative impacts such as environmental damage and price rise, which had a restraining effect on farmers' income growth [51]. There are also contradictory conclusions in the study of the income growth effect of the integrated development of agriculture and tourism through the micro-survey data. Yang (2012) conducted a correspondence analysis based on a survey of 1850 farmers in Chengdu, revealing that income derived from integrated agricultural and tourism facilities, like agritainment, significantly contributes to farmers' income growth [52]. Yao et al. (2016) conducted a study involving 605 farmers in Sichuan, Zhejiang, and Hunan provinces. Their research indicated that annual income for those not involved in rural tourism was 70,000 RMB lower compared to those who participated in such activities [53]. In contrast, Alex et al. (2014) used survey data from 74 rural tourism businesses across 19 villages in Greece's Corinth Mountain region. Their findings highlighted economic losses associated with the development of leisure agriculture in the region. This was primarily due to the dominance of foreign operators in services like bed and breakfast establishments, limiting local farmers' ability to leverage these opportunities for poverty alleviation and income growth [54]. Furthermore, despite the introduction of rural tourism in the case of Pujon Kidul Tourism Village in Indonesia, it resulted in increased employment opportunities but still yielded low economic income. This limitation is attributed to factors related to social capital, particularly the level of education (Nugraha et al., 2021) [55].

The disparities in the conclusions drawn from the aforementioned literature primarily stem from differences in research samples and limitations in research methods. In terms of research samples, the integrated development of agriculture and tourism maybe different in different regions and cultural backgrounds. Most studies adopted provincial level data and ignore regional heterogeneity factors. In terms of research methods, Huang et al. (2022) pointed out that most studies are based on macro data, use farmers' per capita income as the explained variable, and directly regress with the integrated development of agriculture and tourism. Without excluding other driving factors affecting farmers' income growth, the net effect of the integrated development of agriculture and tourism on farmers' income could not be accurately identified [56]. Moreover, the existing studies often neglect other crucial assessment indicators required for the selection of demonstrative counties, such as regional economic development and tourism resources. This oversight introduces a "self-selection" problem during sample selection, which can lead to biased regression results.

Guangxi's integrated development of agriculture and tourism is driving towards the direction of industrialization and specialization (Meng et al., 2021) [57]. However, there are still questions on its impact assessments, such as whether there is an income growth effect, and how to adopt supportive measures with respect to local conditions. Therefore, this paper utilize a county-level panel data and the PSM-DID method to deal with the potential sample self-selection problem and endogenous issue caused by missing variables,

and try to find answers to the above questions by revealing the impacts of the integrated development of agriculture and tourism on farmers' income growth and accounting for its action mechanism.

## 3. Theoretical Analysis and Research Hypotheses

From the perspective of growth pole theory, the integration of agriculture and tourism can be defined as an economic growth pole centered on a specific scale rural area (Rothbarth, 1941) [58]. By attracting the input of external tourists and resources, the integration mode of agriculture and tourism promotes the development and upgrading of local agriculture, forming a new economic growth pole, which can form a strong driving factor and radiation effect, bring more employment opportunities to rural areas, and help farmers achieve income increase. At the same time, according to the Petty–Clark theorem, with economic development, people tend to shift from the primary industry (agriculture) to the secondary industry (manufacturing industry) and the tertiary industry (service industry) (Clarkcg, 1957) [59]. Thus, as far as rural areas are concerned, creating the integration form of agriculture and tourism can promote the development of local tourism and service industries, provide more employment opportunities, and create more economic income. Furthermore, with the development of tourism, the agricultural and tourism integration practices can provide a platform for local farmers to sell agricultural products and other sideline products, and improve their income level. Moreover, according to the Petty–Clark theorem, the higher the proportion of the workforce employed in high-paying industries, the higher the per capita income obtained (Perroux, 1950) [60]. Therefore, the integration of agriculture and tourism can increase the proportion of local farmers engaged in the secondary and tertiary industries, promote the development of local economy, and promote the improvement of rural living standards. Based on the above analysis, Hypothesis 1 is proposed.

**Hypothesis 1 (H1).** *Agriculture and tourism integrated development can promote farmers' income growth.*

### 3.1. The Transmission Mechanism of the Integrated Development of Agriculture and Tourism on Farmers' Income Growth

3.1.1. Increase Rural Employment Opportunities and Farmers' Income

Studies have shown that the integrated development of agriculture and tourism is conducive to promoting the transfer of rural surplus labor to non-agricultural employment (Liu et al., 2017) [26]. On the one hand, the integration of agriculture and tourism helps to promote the development of rural tertiary industry, create more local jobs, and increase the employment opportunities of rural surplus labor force (Richard, 2013) [61]; on the other hand, the integrated development of agriculture and tourism accelerates the process of sending industrial and commercial capital to the rural areas, promotes the multi-sectoral allocation of rural labor resources, and improves the allocation of rural employment structure (Su et al., 2019) [62]. In the reality that non-agricultural wages are constantly higher than agricultural productive wages, multi-sector employment of rural household labor is more conducive to maximizing household income (Schultz, 1964) [63], which leads to a large number of rural labor migrating to cities, resulting in serious problems of rural hollowing and aging. Agriculture and tourism integrated development practices can solve the problem of non-agricultural employment demand of rural labor and its time and space mismatching, which is conducive to attracting talents to return to the rural area. At the same time, due to the return of talents and land rent advantages, in recent years, the integration of agriculture and tourism has created many innovative business models such as "village scenic spots" and "enterprises + farmers", which has broadened the rural employment horizon and entrepreneurial options (Wang and Fan, 2006) [64]. Accordingly, Hypothesis 2 is proposed.

**Hypothesis 2 (H2).** *Agriculture and tourism integrated development can create more rural employment opportunities and farmers' income by strengthening rural non-agricultural employment levels.*

3.1.2. Promote Farmers Transformation and Improve Their Skills

Land scaling management is the foundation for the high-quality development of the integration of agriculture and tourism (Wang et al., 2016) [65]. Before adopting the integrated development of agriculture and tourism, the formation of rural areas was a single, unplanned, and spontaneous behavior. Rural areas and farmland were still maintained in a self-sufficient and fragmented production mode by farmers. However, farmers' income structure began to diversify after introducing those practices. When the tourism service income was greater than the agriculture production income, some rural talents began to carry out a unified land transfer for small farmers in pursuit of greater economic effects, realized large-scale land management by adopting joint management and enterprise management, converted traditional fragmented farmland into a large-scale land setting, provided a better environment for agricultural machinery usage, and improved the mechanization level of agricultural production (Woods, 2009) [66]. As the main body of promoting agricultural science and technology, the government will also strongly support the integrated development of rural areas and create conditions for the production of agricultural science and technology (Barrett and Carter, 2010) [67]. The reason is that the capital accumulation brought by simple agricultural labor cannot meet the conditions for farmers to use advanced agricultural technology and equipment, which blocks the progress of agricultural technology. However, the integrated development of agriculture and tourism can improve farmers' capital income and reduce the capital constraint of agricultural production. Especially for households with a labor shortage, mechanical input can be increased to make up for the insufficient input of labor factors. It not only promoted the progress of agricultural science and technology, but also increased the farmers' income (Gao and Wu, 2017) [68]. Meanwhile, investors' management concept and quality brought by the agriculture and tourism integrated development practices are higher, which can spread advanced production and management knowledge to the rural areas, improve the overall quality of rural agriculture and tourism practitioners, and promote the improvement of labor productivity. Therefore, Hypotheses 3 is proposed.

**Hypothesis 3 (H3).** *The agriculture and tourism integrated development can promote farmers' income growth by improving the use of agricultural technology and agricultural production efficiency.*

*3.2. The Moderating Mechanism of the Integrated Development of Agriculture and Tourism on Farmers' Income Growth*

The integrated development of agriculture and tourism can promote the level of non-agricultural employment and agricultural production efficiency in rural areas, and it can help to improve farmers' income level. However, those practices must be based on the existing rural tourism resource endowment, and the level of regional economic development is also an important factor affecting the income growth. Rural tourism resources are mainly divided into three types: natural scenery, folk culture, and characteristic products. Different types of resources will have different impacts on the agricultural and tourism integrated development practices. In addition, even with enriched rural tourism resources, rural tourism is still difficult to develop without fully equipped infrastructure and complete public services (Mwesiumo et al., 2022) [69]. As "suburban rural tour" has become the preferred way for the public to hang out during the weekend, rural tourism spots around big cities are more favored (Chen et al., 2009) [70]. Due to the insufficient consumption demand in less developed areas, the income growth effect of agriculture and tourism integrated development practices are still difficult to be fully played (Liu et al., 2020) [71]. Thus, this paper further proposes the following Hypotheses 4 and 5.

**Hypothesis 4 (H4).** *The level of economic development positively regulates the promotion effect of the agriculture and tourism integrated development on farmers' income growth.*

**Hypothesis 5 (H5).** *The rural tourism resources endowment positively adjusts the promotion effect of the agriculture and tourism integrated development on farmers' income growth.*

Our research framework is constructed in Figure 1:

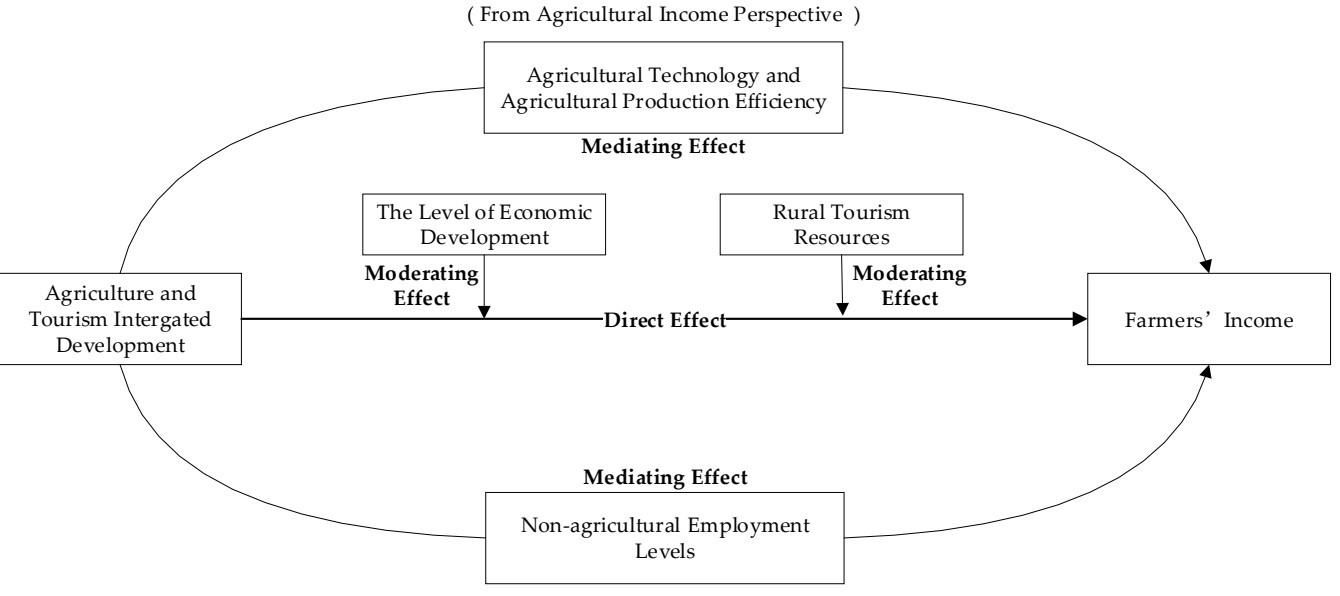

**Figure 1.** The mechanism framework of agriculture and tourism integrated development on farmers' income growth.

## 4. Data and Methods

*4.1. Data Source and Study Area Selection*

4.1.1. Data Source

We used regional county-level data of Guangxi province in this study. Considering the differences in resident composition between urban districts and counties, we first excluded all urban districts and obtained a sample that contains 72 counties within Guangxi. Concerning national leisure agriculture and rural tourism, demonstrative counties were selected from 2010 to 2017; the statistical caliber for the farmers' per capita disposable income has changed since 2005, and with the interference of COVID-19, we then bound the time interval of our dataset from 2005 to 2020. After adopting a PSM method to match control groups for demonstrative counties in the common support area, our final dataset contains 13 counties in the treatment group and 59 counties in the control group, The spatial layout of these research sample is shown in Figure 2. Except for the policy data of demonstrative counties disclosed by the Ministry of Agriculture and Rural Affairs of China, the data of control variables, mediating variables, moderating variables, and covariates were all derived from the Guangxi Statistical Yearbook, Guangxi County Statistical Bulletin, and the CSMAR county economy database. A multiple fitting value method is used to deal with a small number of missing values. In order to eliminate the influence of extreme values, we winsorized the data at 1% level.

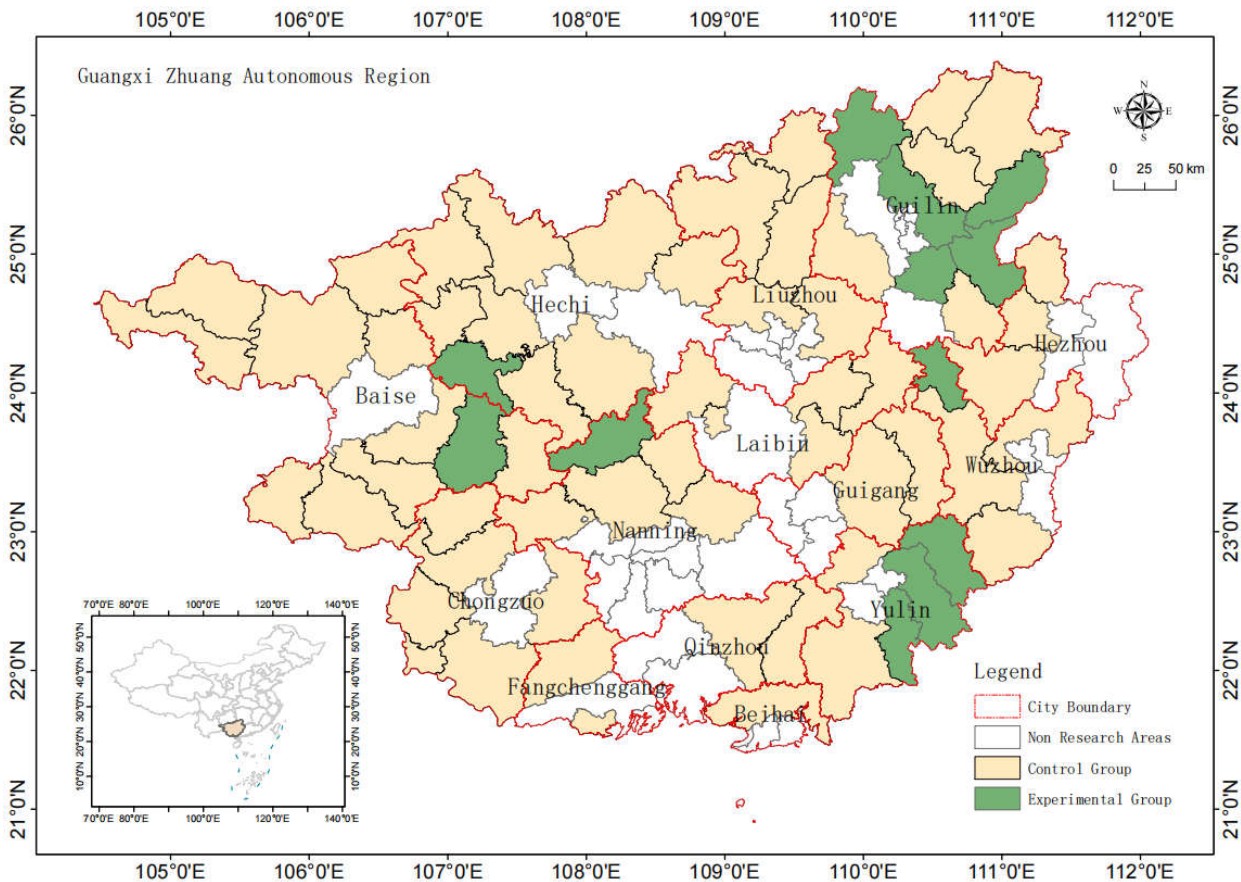

**Figure 2.** Map of the study area. Source: This map is based on the standard map GS(2022)4305 [72] downloaded from the website of the standard map service system of the Ministry of Natural Resources, and the base map is not modified. The following figures are the same.

### 4.1.2. Study Area Selection

Guangxi is located in southwestern China and enriched in natural scenery and tourism resources, such as karst landscapes, which offers unique advantages for the integrated development agriculture and tourism. Figure 3 shows the rural per capita income and tourist attractions spatial distribution map. With its abundant natural and cultural resources, favorable geographical locations, and strong poverty alleviation policies support to improve rural infrastructure, Guangxi's tourism has played an important role in boosting rural income in recent years through the integrated development of agriculture and tourism practices, which provides a good foundation for testing such integrated development impact on farmers' income growth. However, in the process of rural economic development in Guangxi, there are still drawbacks in terms of the overall level of integrated development of agriculture and tourism, and the efficiency of natural and human resources utilization (Qiao and Wu, 2020) [73]. Compared with the eastern provinces, Guangxi's rural per capita income is still significantly lower. Therefore, studying the effect of Guangxi's integrated development of agriculture and tourism on farmers' income growth and its mechanism can help us to depict a better picture for optimizing the industrial policy effects and promoting the rural revitalization in southwestern China.

### 4.1.3. Defining the Integrated Development of Agriculture and Tourism

In China, to facilitate the integrated development of agriculture and tourism, the Ministry of Agriculture and the National Tourism Administration have jointly carried out the establishment of National Leisure Agriculture and Rural Tourism Demonstrative Counties and introduced a series of supporting policies since 2010. Under the support of

these policies, the integrated development of agriculture and tourism has triggered the development of related rural industries in China. A selected county must possess specific characteristics such as the necessary resource endowment, strategic location, industrial characteristics, and cultural history, and take leisure agriculture and rural tourism as the leading industries for its economic development (Zhu, 2022) [74]. Since the demonstrative counties serve as quintessential distinctive examples of China's the integrated development of agriculture and tourism, offering insights into the context of rural revitalization (Hu and Zhong, 2019) [32]. Therefore, we chose it to reveal the main interest of this study.

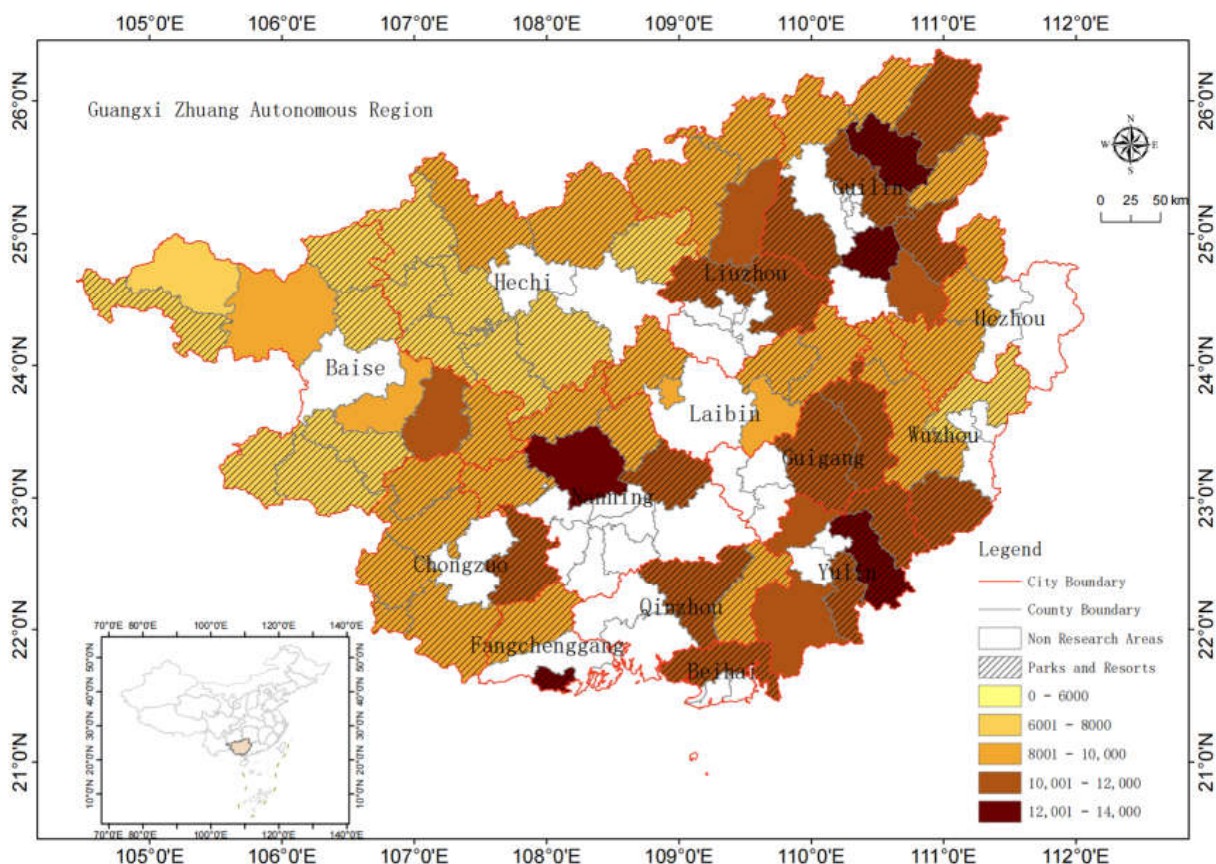

**Figure 3.** Map of rural per capita income and tourist attractions in Guangxi.

*4.2. Model Specification*

This paper takes the integrated development of agriculture and tourism as a "quasi-experiment" and uses the multi-stage DID method to estimate its effect on farmers' income growth. Since we use the selection of National Leisure Agriculture and Rural Tourism Demonstrative Counties as approximate replacement for the implementation of integrated development of agriculture and tourism, one cannot avoid a selection bias problem if simply using non-selected counties as the control group due to demonstrative county selection not being a random selection process. Therefore, we first use PSM method to match the treatment group with similar control group and then selected the successfully matched samples as the regression sample interval, and we adopted the multi-stage DID method to test the net effect of the integrated development of agriculture and tourism on farmers' income.

4.2.1. PSM Modeling

The propensity score matching method (PSM) was first proposed by Heckman. Under the counterfactual framework, the PSM model creates a random simulation experiment, introducing covariates with the same characteristics of the treatment group and the control

group. As long as the propensity score is the same, the treatment group and the control group can be matched. The tendency index can be expressed as the probability of whether the treatment group is processed or not. Direct matching may cause the "dimensional curse" problem, if there are fewer covariates, it may lead to the matching of unsuitable control group, but if there are too many covariates, high-dimension matching will lead to the data sparsity problem. By referring to Rosenbaum (1983) and Imbens (2014) [75,76], we use the caliper nearest neighbor matching method to determine the matching covariates $X_i$. The logit model for estimating propensity score is set as follows:

$$\ln(p_i/1 - p_i) = \alpha_0 + \alpha_1 \text{Base} + u_i \tag{1}$$

$$P(X_i) = \Pr[G_i = 1|X_i] = \exp(X_i\beta)/1 + \exp(X_i\beta) \tag{2}$$

Equation (1) represents the propensity score of each county, and "Base" stands for covariates. The specific process is as follows: we first introduce the control variables into the propensity score estimation equation, and then we bring in the application package of the demonstrative counties covariates to the equation successively to test the likelihood statistic with the benchmark equation and obtain the log-likelihood statistic. Comparing the maximum log-likelihood statistic with the specified threshold, if the statistic is greater than the threshold value, the corresponding covariates is added to the equation, and the above process is repeated. Through matching, samples within the common support domain are selected for the subsequent model test, and samples outside the common support domain are eliminated.

### 4.2.2. Benchmark Regression

As mentioned above, the selection process for National Leisure Agriculture and Rural Tourism Demonstrative Counties is carried out in stages and batches. Due to variations in the establishment time for different demonstrative counties, and the nature of "quasi-experiment", which provides a testable foundation to allow us to capture the differences simultaneously at both the time and regional levels by adopting the "Difference-in-Difference, DID" approach. Additionally, considering the potential for selection bias, we follow the Zhu (2022) [74] and Ge (2023) [77] methods to employ the PSM-DID estimation to assess the net effect of the integrated development of agriculture and tourism on farmers' income growth. The benchmark regression is defined as Equation (3):

$$Y_{it} = \alpha + \beta \text{DID}_{it} + \delta \text{control}_{it} + \lambda_i + \mu_t + \varepsilon_{it} \tag{3}$$

where $Y_{it}$ is farmers' income of county i in time t; $\text{DID}_{it}$ is the core explanatory variable, which is composed of the interaction term between the county dummy variable and the time dummy variable; $\text{control}_{it}$ includes all control variables and covariates; and $\lambda_i$, $\mu_t$, and $\varepsilon_{it}$ represent the individual fixed effect, year fixed effect, and random error terms, respectively.

### 4.2.3. Transmission Mechanism Modeling

According to the transmission mechanism discussion above, the integrated development of agriculture and tourism practices can create more rural employment opportunities and increase farmers' income by strengthening rural non-agricultural employment levels and promoting farmers' income growth by improving the use of agricultural technology and agricultural production efficiency. In order to verify Hypothesis H2 and H3, we use the stepwise regression method by referring to Baron and Kenny (1986) and Judd and Kenny (1981) [78,79] to test the mediating effect, and the model is constructed as follows:

$$Y_{it} = \alpha + \omega_1 \text{DID}_{it} + \delta \text{control}_{it} + \lambda_i + \mu_t + \varepsilon_{it} \tag{4}$$

$$M_{it} = \alpha + \alpha_1 \text{DID}_{it} + \delta \text{control}_{it} + \lambda_i + \mu_t + \varepsilon_{it} \tag{5}$$

$$Y_{it} = \alpha + \beta_1 DID_{it} + \sigma_1 M_{it} + \delta control_{it} + \lambda_i + \mu_t + \varepsilon_{it} \tag{6}$$

where $M_{it}$ is the mediating variable which represents the level of agricultural technology adoption (tech), non-agricultural employment level (n.ag_emp), and agricultural labor productivity (ag_labor), respectively. The total effect of the integrated development of agriculture and tourism is $\omega_1$, the direct effect is $\beta_1$, and the indirect effect is $\alpha_1*\sigma 1$. If part of the effect of the integrated development of agriculture and tourism is indeed generated by the mediating variable $M_{it}$, the following two conditions must be met simultaneously:

a.  $\alpha_1$ has a significant positive influence on the mediating variable $M_{it}$;
b.  In Equation (6), the mediating variable $M_{it}$ has a significant positive impact on farmers' income, and the regression coefficient of $DID_{it}$ decreases after adding the mediating variable; that is, $\beta_1 < \omega_1$.

*4.3. Variable Selection*

4.3.1. The Explained Variable and Explanatory Variable

In order to analyze the impact of agriculture and tourism integrated development on farmers' income, we follow Yang et al. (2022) and Peng et al. (2022) to use the per capita disposable income of rural households as the explained variable [80,81]. The core explanatory variable is the integrated development of agriculture and tourism, which reflects the integration of agriculture and tourism to achieve the goal of coordinated development from economic, ecological, and social aspects.

The selection of National Leisure Agriculture and Rural Tourism Demonstrative Counties is a nationwide supportive policy from the Ministry of Agriculture and Rural Affairs that aims to expand the multiple functions of agriculture and promote high-quality development of rural industries; the policy implements the rural leisure tourism promotion plan, develops multiple functions of agriculture, taps the multiple values of rural areas, and promotes the moderate concentration of resources. The selection criteria also emphasizes the coordination and integration of agriculture and tourism in terms of economy, ecological environment, tourism infrastructure, tourism products and agricultural production conditions, which can better represent the integration level of agriculture and tourism. We utilize this to set up a quasi-natural experiment following Ren et al. (2022) and Huang et al. (2022) [56,82], and we define the integrated development of agriculture and tourism as a dummy variable. If the sample county is selected as a rural tourism demonstrative county, the value is equal to 1, and otherwise it is 0. We then define the period dummy variable and set the year after the sample county is selected as the experiment period and give it a value of 1, and we define the year prior to selection as the control period, with a value of 0. By multiplying the selection and year dummies, the interaction term (did) is obtained, which is the core variable that we concerned with and which indicates the real effect of the experimental group after vigorously promoting the implementation of the integrated development of agriculture and tourism practices. With this setting, the net effect of the integrated development of agriculture and tourism on farmers' income can be better captured.

4.3.2. Mediating Variables and Moderating Variables

The mediating variables and moderating variables we selected for this study are the level of agricultural technology adoption (tech), non-agricultural employment level (n.ag_emp), agricultural labor productivity (ag_labor), economic development level (gdp), and tourism resources (tour), respectively. Among them, the non-agricultural employment level is based on the measurement method given by Andaleeb and Sumit (2020), and Luan et al. (2017) [83,84], using the total number of rural employment and subtracting the rural employment of agriculture, forestry, husbandry, and fishery. The level of agricultural technology adoption refers to Wang et al. (2017) and Zhao et al. (2022) [85,86], taking the ratio of the mechanized cultivation land area over the total cultivated land area as

the standard measurement. In empirical studies, per capita output value is often used to measure labor productivity. We follow Xin and Qin (2020) and use the ratio of the added value of the primary industry at the county level over the number of employees in agriculture, forestry, animal husbandry, and fishery to measure it with the consideration of the county level data availability [87]. The moderating variables economic development level and tourism resources are measured by gross domestic product and whether or not the county has national 4A-level tourist attractions, respectively (Huang et al., 2022) [56].

### 4.3.3. Control Variables and Covariates

In order to exclude other influences on farmers' income, based on the practices of scholars Ren et al. (2022) and Huang et al. (2022) and the availability of data, we include agriculture development level (ag_level), industrial structure (ind), fiscal expenditure (fiscal_ex), finance development level (fin), fixed asset investment (invest), education input level (edu), communication infrastructure situation (comm), and mechanization level (mech) into the model as control variables set. For covariates set, we first include all control variables as the base covariates set [56,82]. Through the application package of demonstrative counties, we found that agricultural population and tourism development level are included as indicators for application. Therefore, we also select rural agriculture employed population (r_ag.pop) rural employed population (rural_pop) and total number of tourists received (tour_reci.) into our covariates set. The descriptive statistical characteristics and the data source of the variables are shown in Table 1.

**Table 1.** The descriptive statistical characteristics and the data source of the variables.

| Variable | Definition | Obs | Mean | Std. Dev. | Min | Max | Data Source |
|---|---|---|---|---|---|---|---|
| Y | farmers' income | 1152 | 0.7131 | 0.4200 | 0.1407 | 2.1173 | Guangxi Statistical Yearbook |
| did | interaction term | 1152 | 0.0755 | 0.2643 | 0 | 1 | Ministry of Ag. and Rural Affairs |
| tech | ag. tech adoption | 1152 | 0.3010 | 0.1963 | 0.0055 | 0.9812 | Guangxi Statistical Yearbook |
| n.ag_emp | non-ag. employment | 1152 | 9.1944 | 9.0531 | 0.4800 | 51.3375 | CSMAR county economy database |
| ag_labor | ag labor productivity | 1152 | 1.3393 | 0.8444 | 0.1955 | 6.6900 | CSMAR county economy database |
| gdp | economic dev. level | 1152 | 82.5900 | 69.5779 | 4.8330 | 372.8099 | Guangxi Statistical Yearbook |
| tour | tour resources dummy | 1152 | 0.3897 | 0.4879 | 0 | 1 | Guangxi County Statistical Bulletin |
| ag_level | agriculture dev. level | 1152 | 0.2795 | 0.0885 | 0.0652 | 0.5168 | CSMAR county economy database |
| ind | industrial structure | 1152 | 0.3361 | 0.0982 | 0.1384 | 0.6946 | Guangxi Statistical Yearbook |
| fiscal_ex | fiscal expenditure | 1152 | 0.2758 | 0.1794 | 0.0591 | 1.2646 | Guangxi Statistical Yearbook |
| fin | finance dev. level | 1152 | 0.5499 | 0.2926 | 0.1087 | 3.0529 | Guangxi Statistical Yearbook |
| invest | fixed asset invested | 1152 | 0.9012 | 0.4639 | 0.0370 | 4.5767 | Guangxi Statistical Yearbook |
| edu | education input level | 1152 | 0.1410 | 0.0361 | 0.0264 | 0.3523 | Guangxi Statistical Yearbook |
| comm | comm. infrastructure | 1152 | 0.5989 | 0.4103 | 0.0364 | 3.0031 | Guangxi Statistical Yearbook |
| mech | mechanization level | 1152 | 31.4210 | 19.2885 | 4.0000 | 129.0000 | Guangxi Statistical Yearbook |
| r_ag.pop | rural ag. employed pop | 1152 | 1.6258 | 1.0231 | 0.2920 | 5.3908 | Guangxi Statistical Yearbook |
| rural_pop | rural employed pop. | 1152 | 2.5452 | 1.8837 | 0.4610 | 10.5245 | Guangxi Statistical Yearbook |
| tour_reci. | tourists received (in mil) | 1152 | 25.7739 | 28.0781 | 1.5258 | 152.0974 | Guangxi Statistical Yearbook |

## 5. Empirical Result Discussion

### 5.1. Average Effect of the Integrated Development of Agriculture and Tourism on Farmers' Income Growth

Since it is impossible to observe the difference in farmer's income between sample counties, this paper first selected rural agriculture employed population (r_ag.pop), rural employed population (rural_pop), total number of tourists received (tour_reci.), and control variables to perform a matching process. The adjacent matching method in PSM matched the data according to a 1:2 matching, with a maximum distance of 0.05 between the test group and the matching group. The difference in differences (DID) estimation for the matched sample was then calculated with a common trend assumption. As seen in Figure 4, the kernel density curves of the treatment group and the control group differed notably before PSM matching. After matching, the kernel density curves of the two groups of samples coincided, proving that the characteristic variables of the sample county areas in the two sample groups were relatively similar.

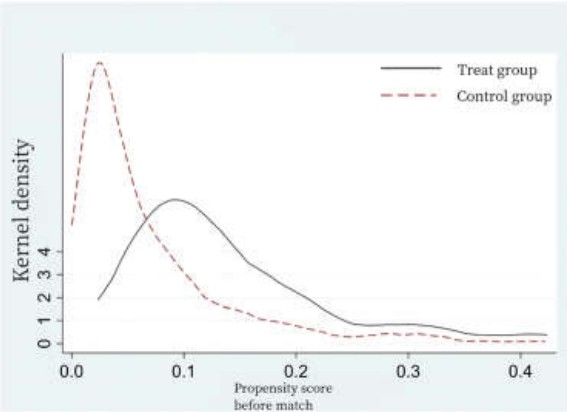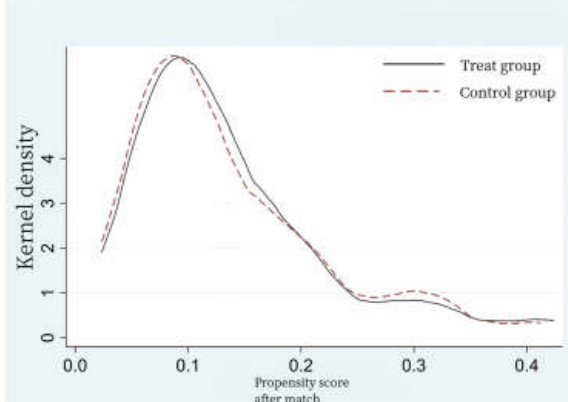

**Figure 4.** Comparison of propensity scores between the treatment group and control group pre- and post-PSM matching.

The matching balance test results of the scores of each covariate are shown in Table 2. The absolute values of standard deviations of all variables show a downward trend, and the t-test results were not significant. It indicates that there is no significant difference between the matching treatment group and the control group, and the PSM balance test is satisfied.

**Table 2.** PSM balance test result.

| Variable | | Mean | | % Bias | t-Test | |
|---|---|---|---|---|---|---|
| | | Treatment | Control | | t | P > t |
| ag_level | Unmatched | 0.2434 | 0.2825 | −48.4 | −3.99 | 0.000 |
| | Matched | 0.2434 | 0.2333 | 12.5 | 0.81 | 0.422 |
| ind | Unmatched | 0.3890 | 0.3319 | 60.3 | 5.28 | 0.000 |
| | Matched | 0.3827 | 0.3841 | −1.5 | −0.09 | 0.928 |
| fiscal_ex | Unmatched | 0.2705 | 0.2762 | −3.6 | −0.29 | 0.772 |
| | Matched | 0.2687 | 0.2892 | −12.9 | −0.99 | 0.326 |
| fin | Unmatched | 0.7052 | 0.5373 | 63.3 | 5.20 | 0.000 |
| | Matched | 0.6977 | 0.7466 | −18.4 | −1.16 | 0.247 |
| invest | Unmatched | 1.0577 | 0.8885 | 41.6 | 3.28 | 0.001 |
| | Matched | 1.0546 | 1.0414 | 3.2 | 0.21 | 0.831 |
| edu | Unmatched | 0.1385 | 0.1413 | −7.1 | −0.67 | 0.501 |
| | Matched | 0.1391 | 0.1371 | 5.3 | 0.35 | 0.724 |
| comm | Unmatched | 0.8677 | 0.5770 | 77.0 | 6.46 | 0.000 |
| | Matched | 0.8564 | 0.8839 | −7.3 | −0.48 | 0.635 |
| mech | Unmatched | 36.5290 | 31.0040 | 31.4 | 2.58 | 0.010 |
| | Matched | 36.5300 | 34.0060 | 14.4 | 0.88 | 0.383 |
| r_ag.pop | Unmatched | 220.0000 | 260.0000 | −19.6 | −1.62 | 0.106 |
| | Matched | 220.0000 | 210.0000 | 7.0 | 0.53 | 0.600 |
| rural_pop | Unmatched | 140.7300 | 164.3600 | −26.1 | −2.07 | 0.038 |
| | Matched | 142.0800 | 133.7200 | 9.2 | 0.69 | 0.490 |
| tour_reci. | Unmatched | 6066.8000 | 2292.3000 | 118.8 | 12.89 | 0.000 |
| | Matched | 5687.3000 | 5453.7000 | 7.4 | 0.40 | 0.689 |

*5.2. Benchmark Regression Result Discussion*

Table 3 presents the estimated results of DID under different methods. The interaction term shows a positive correlation at 1% level of significance in all models, indicating that the demonstrative counties focusing on the integrated development of agriculture and tourism have a promotion effect on farmers' income growth in comparing with non-demonstrative counties, which is consistent with Ren et al.'s (2022) conclusion [82]. Thus, Hypothesis 1 is confirmed. Compared with models (1) and (2), when the individual fixed effect and the time fixed effect were not added, farmers' income in the treatment group increased

by 1660 RMB on average, when compared with the control group, while farmers' income growth dropped to 373 RMB after the individual fixed effect and the time fixed effect were considered. After PSM matching, the approximate random matching group is more similar to the quasi-natural experiment, and farmers' income growth increased to 439 RMB (model 3), indicating that with the continuous robustness of the model design, the net promotion effect of the integrated development of agriculture and tourism on farmers' income growth is also increasing, which is consistent with the dynamic regression results shown in Table 4.

**Table 3.** Benchmark regression result.

| Variable | Y | | |
| --- | --- | --- | --- |
| | OLS (1) | FE (2) | PSM-DID (3) |
| did | 0.1660 *** | 0.0373 *** | 0.0439 *** |
| | (3.3334) | (4.0238) | (4.4151) |
| Control | YES | YES | YES |
| Individual fixed | NO | YES | YES |
| Time fixed | NO | YES | YES |
| N | 1152 | 1152 | 981 |
| Adj. $R^2$ | 0.8160 | 0.9812 | 0.9804 |

Note: *** $p < 0.01$. The value in parentheses below the coefficient is the standard error.

**Table 4.** Parallel trend test result.

| Variable | Y |
| --- | --- |
| | Parallel Trend Test |
| pre4 | −0.0217 |
| | (−0.86) |
| pre3 | −0.0117 |
| | (−0.46) |
| pre2 | −0.00371 |
| | (−0.15) |
| pre1 | 0.0184 |
| | (0.72) |
| current | 0.0357 |
| | (1.45) |
| post1 | 0.0361 |
| | (1.47) |
| post2 | 0.0506 ** |
| | (2.05) |
| post3 | 0.0570 ** |
| | (2.32) |
| post4 | 0.0504 * |
| | (1.73) |
| _cons | 0.775 *** |
| | (261.30) |
| Individual fixed | YES |
| Time fixed | YES |
| N | 981 |
| Adj. $R^2$ | 0.9624 |

Note: *** $p < 0.01$; ** $p < 0.05$; * $p < 0.1$. The value in parentheses below the coefficient is the standard error.

*5.3. Robustness Checks*

5.3.1. Parallel Trend Test

The primary premise of multi-stage DID is that the treatment and control groups need to maintain a common trend before the experiment occurs. Therefore, this paper uses Beck's (2010) processing method to test the parallel trend, and it multiplies the year dummy

variable before the demonstrative county is selected by the county dummy variable as the explanatory variable [88]. The specific setting is as follows:

$$Y_{it} = \partial_0 + \sum_{d=-4}^{-1} \beta_d pre_d + \beta_0 current + \sum_{s=1}^{4} \beta_s post_s + \lambda_i + \mu_t + \varepsilon_{it} \tag{7}$$

The significance of $pre_d$ and $post_s$ is the main focus of parallel trend test, which reflects the difference in time trend between the treatment group and the control group. $pre_d$ represents the 1 to 4 years before the integrated development of agriculture and tourism practices, current represents the year that implementing integrated development of agriculture and tourism practices, and $post_s$ focuses on the next 4 years after the integrated development of agriculture and tourism practices. For any time period longer than 4 years, we assign either d = 4 (if before), or s = 4 (if after), respectively, and current is set as the base year. Table 3 shows the estimated results of Equation (4). One can clearly see that in the first 4 years before the integrated development of agriculture and tourism practices, all estimated coefficients are not significant, but after the practices are implemented, except for the first year, other variables were positively significant with farmers' income, and the income growth effect coefficient showed an increasing trend. The results indicate that the benchmark regression model meets the parallel trend hypothesis, and the integrated development of agriculture and tourism helps farmers on gaining more income.

### 5.3.2. Replace Explained Variables

In the benchmark regression, the explained variable uses the nominal income data directly provided by the Guangxi Statistical Yearbook without taking into account the impact of price and inflation factors. Thus, we replaced farmers' income data to real income $Y_2$ (2005 as the base period) eliminating the impact of price factors caused by economic fluctuations. The re-estimated results are shown in models (4)–(6) in Table 5. The interaction term is still significantly positive in all three models, which indicates that the empirical results showing that the integrated development of agriculture and tourism effectively improves farmers' income growth was robust.

**Table 5.** Robustness check results.

| Variable | | $Y_2$ | | Y |
|---|---|---|---|---|
| | OLS (4) | FE (5) | PSM-DID (6) | PSM-DID (7) |
| did | 1.0727 *** | 0.2488 *** | 0.2791 *** | 0.0205 * |
| | (3.3394) | (4.2091) | (4.3887) | (1.9559) |
| Control | YES | YES | YES | YES |
| Individual fixed | NO | YES | YES | YES |
| Time fixed | NO | YES | YES | YES |
| N | 1152 | 1152 | 981 | 826 |
| Adj. $R^2$ | 0.8176 | 0.9807 | 0.9794 | 0.9803 |

Note: *** $p < 0.01$; * $p < 0.1$. The value in parentheses below the coefficient is the standard error.

### 5.3.3. Matching Method Switch

The caliper nearest neighbor matching method is used in propensity score matching process. In order to reduce the impact of limitations of matching methods on empirical results, we introduced the kernel matching method (Bockerman and Ilmakunnas, 2021) and re-performed a matching process [89]. The regression results after matching are shown in model (7) in Table 5. Although the number of the observations decreased to 826, the estimated the coefficient of the interaction term still showed a positive correlation at a 10% significance level, and the control variables were tended to be consistent with previous regression results. Thus, the conclusion was not affected by changing the propensity score matching method.

5.3.4. Placebo Test

According to the existing literature, there are various factors affecting the effect of farmers' income growth on the integrated development of agriculture and tourism. Although we mentioned, tested, and verified the assumptions of the PSM-DID in above analysis, there may still be pseudo-regression results caused by non-observed missing variables. Therefore, we followed Heckman and Ichimura (1998) to execute a placebo test [90]. The detail process is as follows: first, 13 counties were randomly selected from 72 counties according to the year, and these counties were set as the "pseudo-experimental group", and the remaining samples were used as the control group; next, an experiment time was randomly generated for each newly generated "pseudo-experimental group"; finally, we regressed the interaction terms of "pseudo-experimental group" and "pseudo-experimental time" to obtain the estimated coefficient of the simulation. In theory, since the interaction terms are randomly generated, there is no significant effect on the explained variables. Therefore, the expected coefficient of the interaction term is 0, and the *p*-value is not significant. We repeated the above random process 500 times, and the results are shown in Figure 5. One can see from the figure that most of the blue dots are distributed above the red virtual horizontal line, indicating that the *p*-value of most of the DID regression coefficients is not significant, and the mean value is close to 0. Meanwhile, the actual estimated coefficient of the red virtual vertical line in the figure is 0.0439, which is obviously an outlier in the random distribution, with only one point on its right side. Therefore, the placebo test verified the benchmark regression results and proved that there was no obvious missing variable bias in the empirical results. The empirical results were robust.

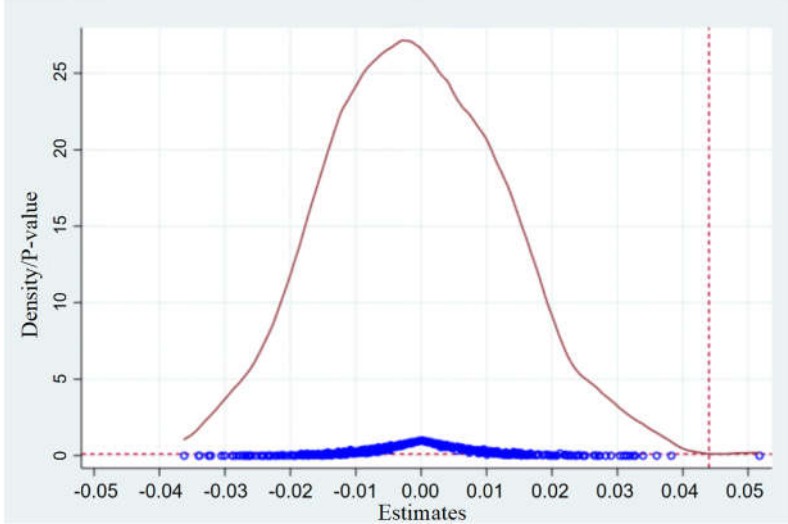

**Figure 5.** The placebo test result. Note: the X-axis represents the DID coefficient estimated based on 500 random selection of 13 counties as the pseudo-experimental group and randomly assigned policy implementation time; the Y-axis represents its corresponding *p*-value and density; the blue dots are the *p*-values of DID after each random matching; the red curve represents the kernel density curve; the red virtual horizontal line is y = 0.1; and the red virtual vertical line is x = 0.0439. The placebo test was programmed using Stata15.1 software.

*5.4. Mechanism Test Result Discussion*

The integrated development of agriculture and tourism is an important support to promote rural revitalization and achieve common prosperity. The regression results of the benchmark model show that the integrated development of agriculture and tourism has significantly promoted farmers' income growth in Guangxi. However, what is the underlying mechanism of the integrated development of agriculture and tourism on farmers' income growth? Recalling the previous sections, the integrated development of agriculture and tourism may promote farmers' income growth through three ways: enhancing rural non-

agricultural employment level to promote rural employment structure change, improving the level of agricultural production technology, and strengthening agricultural production efficiency. Meanwhile, due to the heterogeneity of regional resource endowment and the non-balanced economic development, tourism resource endowments and economic development level may moderate the promoting effect of integrated development of agriculture and tourism on farmers' income growth. Thus, this paper also tested the mechanism, and the results are discussed as follows:

5.4.1. Mediating Effect Test

Table 6 shows the results of the test on the rural non-agricultural employment level. Among them, model (8) is the baseline regression result, and model (9) shows the direct effect of the integrated development of agriculture and tourism on the level of non-agricultural employment. However, the model result is contrary to the assumed expectation, and the demonstrative counties focusing on the integrated development of agriculture and tourism significantly inhibit the non-agricultural employment level. When simultaneously including the interaction term and "n.ag_emp" into the regression equation, the result (model 10) is consistent with our expectations. The increase in non-agricultural employment level positively promotes the farmers' income growth, and the estimated coefficient of the interaction term also increases, which indicates that there is a masking effect between the integrated development of agriculture and tourism and the non-agricultural employment level. The specific explanation is that the integrated development of agricultural and tourism inhibits the non-agricultural employment level, and the non-agricultural employment level has a positive impact on farmers' income growth. This is the opposite of Hypothesis 2.

**Table 6.** Mediating effect test results for enhancing rural non-agricultural employment level to promote rural employment structure change.

| Variable | (8) Y | (9) n.ag_emp | (10) Y | (11) r_ag.pop | (12) Y |
|---|---|---|---|---|---|
| n.ag_emp | | | 0.0137 *** | | |
| | | | (12.04) | | |
| r_ag.pop | | | | | −0.0120 *** |
| | | | | | (−10.46) |
| did | 0.1280 *** | −2.0790 *** | 0.1560 *** | −29.9100 *** | 0.1200 *** |
| | (6.22) | (−3.54) | (6.49) | (−4.92) | (6.05) |
| Control | YES | YES | YES | YES | YES |
| N | 981 | 981 | 981 | 981 | 981 |
| Adj. $R^2$ | 0.9128 | 0.6888 | 0.8140 | 0.7315 | 0.8079 |

Note: *** $p < 0.01$. The value in parentheses below the coefficient is the standard error.

The mediating effect test results for improving the level of agricultural production technology and strengthening agricultural production efficiency are shown in Table 7. Models (13) and (14) represent the direct effect and indirect effect of the level of agricultural production technology. The results of model (13) show that the integrated development of agriculture and tourism has a significant promotion effect on the adoption of agricultural machinery technology, and when the interaction term and agricultural production technology level are added to model (14) at the same time, both significantly promote the farmers' income growth, with an estimated coefficient of interaction term decrease. The results show that the level of agricultural production technology is served as an intermediary mechanism of the integrated development of agriculture and tourism to promote the farmers' income growth. Similarly, the results of model (15) and (16) also prove that the improvement of agricultural production efficiency is another important intermediary mechanism. Therefore, the above evidence proves that the integrated development of agriculture and tourism can promote farmers' income growth through improving the level of agricultural production technology and agricultural production efficiency, and thus, Hypothesis 3 is verified.

**Table 7.** Mediating effect test results for improving the level of agricultural production technology and strengthening agricultural production efficiency.

| Variable | (8) Y | (13) tech | (14) Y | (15) ag_labor | (16) Y |
|---|---|---|---|---|---|
| ag_labor | | | | | 0.2230 *** |
| | | | | | (22.53) |
| tech | | | 0.2960 *** | | |
| | | | (7.48) | | |
| did | 0.1280 *** | 0.1350 *** | 0.0880 *** | 0.1730 *** | 0.0890 *** |
| | (6.22) | (8.00) | (4.26) | (3.13) | (5.40) |
| Control | YES | YES | YES | YES | YES |
| N | 981 | 981 | 981 | 981 | 981 |
| Adj. $R^2$ | 0.9128 | 0.7335 | 0.9126 | 0.8557 | 0.9442 |

Note: *** $p < 0.01$. The value in parentheses below the coefficient is the standard error. The following tables are the same.

Specifically, the integrated development of agriculture and tourism will lead to the large-scale development of local land, which provides a good foundation for the large-scale operation of agricultural technology, and the large-scale adoption of agricultural technology promotes the efficiency of agriculture and increases farmers' agricultural productive income. At the same time, the large-scale adoption of agricultural technology reduces the cost of agricultural production and human capital input, promotes the large-scale operation of agriculture, improves the efficiency of agricultural production, and increases farmers' income.

5.4.2. Moderating Effect Test

In order to test the moderating mechanism of tourism resource endowment on the income growth effect of the integrated development of agriculture and tourism, we selected national 4A-level tourist attractions as dummy variables to represent tourism resource endowment. In this study, we divided the full sample into two groups according to whether each county has national 4A-level scenic attractions for a hierarchical regression (shown in Table 8). According to the regression results of model (17) and (18), although the interaction term in both models significantly promoted the farmers' income growth, one can see that the integrated development of agriculture and tourism had significantly different impacts on farmers' income in regions with different tourism resource endowments. The income growth effect in tourist areas with a 4A level or above rating is twice that of non-4A-level scenic attractions. Therefore, H4 is verified, such that the more rural tourism resources there are, the more significant the income growth effect of the integrated development of agriculture and tourism will be. This may be explained by the fact the relatively good infrastructure conditions and large tourist source market of 4A-level tourist attractions, which play a strong driving role in the integration and development of agriculture and tourism practices (Grunwell and Ha, 2020) [91].

**Table 8.** Moderating effect test results.

| Variable | Tour = 1 Y (17) | Tour = 0 Y (18) | High GDP Y (19) | Low GDP Y (20) |
|---|---|---|---|---|
| did | 0.0400 *** | 0.0240 * | 0.0360 *** | 0.0230 |
| | (2.84) | (1.77) | (3.24) | (1.48) |
| Control | YES | YES | YES | YES |
| Individual fixed | YES | YES | YES | YES |
| Time fixed | YES | YES | YES | YES |
| N | 405 | 566 | 362 | 604 |
| Adj. $R^2$ | 0.9834 | 0.9879 | 0.9908 | 0.9708 |

Note: *** $p < 0.01$; * $p < 0.1$. The value in parentheses below the coefficient is the standard error.

From a practical point of view, even if rural tourism resources are very rich, it is difficult for leisure agriculture and rural tourism to develop without perfect infrastructure and public services, and the state of infrastructure and public service level depend on the level of local economic development to a large extent. Therefore, we divided the full sample into high and low groups according to the average mean of the GDP variables of each county for testing. The regression results are shown in model (19) and (20) in Table 8. We found that there are significant differences in the impact of the integrated development of agriculture and tourism on farmers' income growth in regions with different economic development levels. The income growth effect of the integrated development of agriculture and tourism can be exerted to a greater extent in the regions with a higher economic level, while it is not significant in the regions with a lower economic level. This verifies H5, such that the higher the level of economic development, the more significant the income growth effect of the integrated development of agriculture and tourism.

## 6. Further Discussion

The above results on the mediating effect of non-agricultural employment show that the integrated development of agriculture and tourism inhibits the level of non-agricultural employment, which is contrary to Hypothesis 2. In order to explain this phenomenon, we recalled the existing scholarly works and adopted a reverse indicator, "the total number of employees in agricultural, forestry, animal husbandry and fishery industries" (Zhang, 2019), to measure the non-agricultural employment level as a replacement [92]. The logic behind this is that since there are fewer industrial enterprises in rural areas, it can be approximately ignored. The decrease in the number of rural workers in agricultural, forestry, animal husbandry, and fishery industries means that the rural employment structure is shifting to non-agricultural employment, thereby improving the level of non-agricultural employment. The results of model (11) and model (12) also show that the integrated development of agricultural and tourism promotes the outflow of rural agricultural employment, and the rural agricultural employment level significantly inhibits the effect of farmers' income growth. By comparing the two mediating effect models in Table 6, we can see that the integrated development of agriculture and tourism does squeeze out some agricultural employees, but these agricultural employees who are squeezed out may not join the non-agricultural sector to obtain higher wage income, resulting in the model (9) result.

Specifically, the reasons for this phenomenon may be concluded in the following three aspects: First, due to regional heterogeneity, the Guangxi's economic development level has fallen relatively behind, especially the economic status of rural areas in the lower reaches of the countrywide. Although Guangxi has advantages in natural tourism resources in rural areas, the investment in the integrated development of agriculture and tourism project is a heavy-asset investment, including the establishment of infrastructure of scenic spots, the construction of sightseeing roads, and the later maintenance costs. Therefore, the rural talents or township governments cannot afford to support such investment alone, but they rely on the external industrial and commercial capitals attracted by the rural revitalization strategy, such as "Jiahua Ecological Orchard", invested in by Malaysia Jiahua Group; "China Eastern Airlines Guangxi Agriculture and Tourism Integration Industrial Base", jointly invested in by China Eastern Airlines and Guangxi Zhuang Autonomous Region Government; "Maite Smart Agriculture Demonstration Park", invested in by Maite precision Machinery (Guangxi) Co., Ltd. (Nanning, China); and other projects. At the same time, the intervention of external industrial and commercial capitals have enhanced the rural land transfer process to meet the needs of project implementation through village community organizations (Zhou, 2015) [93]. In this process, most farmers with fragmented land will choose to transfer their land to gain a one-time land rent capital income (far higher than their productive income). After losing land usage rights, some farmers are not able to join the tourism industry chain to obtain non-agricultural wage income (due to lack of employment skills matching rural tourism) and forced to leave the rural employment environment for other working options. Second, in order to pursue better employment

opportunities, educational resources, medical facilities, and income levels in the cities, some farmers sell their land to investors for the integrated development of agricultural and tourism project to accumulate capital and migrate to the city. Third, since there are no data of rural non-agricultural employees in the statistical yearbook, in our study, the number of rural non-agricultural employees is obtained by subtracting the number of agricultural, forestry, animal husbandry, and fishery employees from a total employed population to approximate the number of rural non-agricultural employees, which does not include part-time farmers and some mobile traders who cannot be included in the statistical standard. It may also lead to a pseudo-regression problem.

However, some scholars have verified the intermediary path mechanism whereby the integrated development of agriculture and tourism can promote the level of non-agricultural employment and thus increase farmers' income by using national demonstrative counties samples. In order to verify whether the regional heterogeneity in southwestern China leads to the contrary results in our study, we added samples from Yunnan and Guizhou province to examine the mediating effect of non-agricultural employment level (shown in Table 9).

**Table 9.** Parallel mediating effect test results for Yunan and Guizhou.

| Variable | Yunan Province | | | Guizhou Province | | |
|---|---|---|---|---|---|---|
| | Y (21) | n.ag_emp (22) | Y (23) | (24) | n.ag_emp (25) | Y (26) |
| did | 1.4930 *** | −11.8400 ** | 1.6520 *** | 0.1720 | 31.1500 *** | 0.0010 |
| | (4.98) | (−2.32) | (4.64) | (0.59) | (4.16) | (0.00) |
| n.ag_emp | | | 0.0040 ** | | | 0.0060 *** |
| | | | (2.15) | | | (4.86) |
| Control | YES | YES | YES | YES | YES | YES |
| N | 1792 | 1792 | 1792 | 1152 | 1152 | 1152 |
| Adj. $R^2$ | 0.5194 | 0.2369 | 0.5204 | 0.6855 | 0.3645 | 0.6940 |

Note: *** $p < 0.01$; ** $p < 0.05$. The value in parentheses below the coefficient is the standard error.

Models (21)–(23) and (24)–(26) represent the total, direct, and indirect effects, respectively, of non-agricultural employment level in Yunnan and Guizhou. In terms of results, the sample of Yunnan Province supports the above analysis results. There is also a masking effect between the integrated development of agriculture and tourism and the level of non-agricultural employment, which shows that the development of the integrated project of agriculture and tourism inhibits the level of non-agricultural employment, and the level of non-agricultural employment has a positive impact on the increase in farmers' income. It is further proved that the inhibitory effect of the integrated development of agriculture and tourism on the level of non-agricultural employment may be caused by regional differences. However, according to the case of Guizhou, the effect of the integrated development of agriculture and tourism is not significant. Combined with the analysis of the moderating effect, the economic development level of Guizhou province is the lowest in western China except Tibet. As a result, the development of agricultural and tourism integration projects is restricted by the level of local economic development, and it cannot form a strong driving force. In terms of tourism resources, the level of tourism resources in Yunnan Province is at the forefront, which can bring a stable tourist market to support the integrated development of agriculture and tourism, while most of the demonstrative counties in Guangxi are located in and around the national tourism destinations, which can also rely on the local unique landscape and scenery to promote the rapid development of agriculture and tourism integration, so as to promote farmers' income growth.

## 7. Conclusions

To investigate the relationship between the integrated development of agriculture and tourism and farmers' income growth in southwestern China, this study conducted a series of empirical analyses using a PSM-DID method with a panel dataset of 72 counties

within Guangxi province from 2005 to 2020. Our findings reveal a significant positive effect of integrated agriculture and tourism development on farmer's income growth, showing a consistent upward trend over time. The mechanism analysis highlights that improvements in agricultural production technology and agricultural production efficiency serves as crucial drivers for increasing farmer's income. Interestingly, we uncovered a masking effect associated with the integrated development of agriculture and tourism and the level of non-agricultural employment in southwestern China. The possible implication is that external industrial and commercial capital investment has displaced income and non-agricultural employment opportunities that farmers initially derived from agricultural work by compensating them with the one-time land rent capital income, which forces them to seek for jobs outside the rural environment. Furthermore, the results from the moderating effect analysis emphasize the importance of considering regional variations in tourism resource endowment and economic development levels when planning and implementing integrated agriculture and tourism development strategies.

This study has identified significant potential in the development of the integration of agriculture and tourism for enhancing farmers' income in southwestern China. However, there are noteworthy suggestions for improvement. Firstly, to effectively promote income growth through the integrated development of agriculture and tourism, government authorities at all levels should judiciously employ policy tools and allocate resources, taking into consideration the local realities. In regions with a robust economic base and abundant tourism resources, governments should leverage financial and resource advantages to establish distinctive local models of integrated agriculture and tourism. Additionally, policymakers should enhance training and provide guidance to rural tourism practitioners, elevating service quality to ensure tourist satisfaction and generating greater benefits for farmers. Simultaneously, governments must actively facilitate the dissemination of agricultural technology, harness the advantages of scale in agricultural production, enhance the quality of agricultural products, and diversify income opportunities for farmers beyond agricultural activities. Secondly, policymakers may focus on to a potential structure change in rural employment when implementing the integrated of agriculture and tourism practices. This should involve empowering farmers to switch between agricultural and non-agricultural sectors. The government should actively carry out agricultural and non-agricultural training programs to help farmers to foster new skills such as rural e-commerce to further enable farmers' capability of boosting their income. Lastly, the sustainability of interests for all stakeholders should be the cornerstone of the successful integration of agriculture and tourism practices.

The results of this study provide a continuous assessment of how the integration of agriculture and tourism can stimulate the growth of farmers' income and bolster rural economic development. Furthermore, it presents a new perspective for evaluating the impact of the integrated development of agriculture and tourism toward the rural revitalization in China. Nonetheless, a masking effect exists between the integrated development of agriculture and tourism and non-agricultural employment level in southwestern China, which inhibits non-agricultural employment opportunities. Therefore, the lesson we learned is that we have to maintain a sustainable course in the process of the integrated development of agriculture and tourism. How to ensure all stakeholders have a fair right to share the benefits and development opportunities, and further promote the sustainable development of agriculture and tourism integration practices, is the direction that is worth more attention in the future.

Although this study examines the impact and mechanism of the integrated development of agriculture and tourism on farmers' income growth, further research is needed on how such integrated development adjusts the rural employment structure.

**Author Contributions:** Conceptualization, Y.L. and T.X.; methodology, Y.L., T.X. and D.M.; validation, A.G. and Y.C.; formal analysis, Y.L. and T.X.; investigation, T.X. and D.M.; data curation, T.X.; writing—original draft preparation, T.X. and A.G.; writing—review and editing, Y.L. and Y.C.; supervision, A.G. and Y.C.; funding acquisition, Y.L. All authors have read and agreed to the published version of the manuscript.

**Funding:** This research was funded by Guangxi Philosophy and Social Science Project (No. 22BJL003) and Pearl River-Xijiang River Economic Belt Development Institute Commissioned Research Project (No. ZX2023002).

**Institutional Review Board Statement:** Not applicable.

**Data Availability Statement:** All the data are obtained from the Guangxi Statistical Yearbook, Guangxi County Statistical Bulletin and the CSMAR county economy database, except for the policy data of demonstrative counties, which were disclosed by the Ministry of Agriculture and Rural Affairs of China. The data are available upon request from the corresponding author.

**Acknowledgments:** The authors acknowledge the support provided by their respective institutions.

**Conflicts of Interest:** The authors declare no conflict of interest.

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
