# Peer review of "Does the Integrated Development of Agriculture and Tourism Promote Farmers’ Income Growth? Evidence from Southwestern China"

_agriculture, doi:10.3390/agriculture13091817_

Round 1
Reviewer 1 Report
-
1. Chapter 1 requires further development in terms of research motivation and gaps in the existing literature. It is essential to elucidate the significance and necessity of this topic to establish its relevance.
-
2. The literature review should be organized with appropriate subheadings to enhance its structure.
-
3. Additional research literature pertaining to the intersection of agriculture and tourism should be incorporated to enrich the content and provide a comprehensive understanding of the subject.
-
4. It's recommended to reconsider the method used to create the research framework. You might find value in referring to other established research frameworks for inspiration and best practices.
-
5. Unfortunately, the author has extensively described the data analysis process, but there is a limited exploration of research recommendations and industry implications in Chapter 4. These aspects should be thoroughly explained to ensure a well-rounded discussion.
-
6. The research lacks a clear discussion of its limitations and future recommendations. It's crucial to acknowledge the study's limitations and offer insights into potential areas for future research.
- 7. The format does not conform to MDPI requirements
Extensive editing of the English language required
Author Response
Dear Reviewer,
We are grateful to your valuable comments and suggestions. Based on your reviewing report, we did a point-by-point rechecks and revisions. A detailed response is attached. We hope that our revised manuscript and response can make you satisfied.
Again, thanks for your time and comments. We look forward to hearing from you.
Sincerely,
Yuxi Luo
Tianren Xiong
Defeng Meng
Anrong Gao
Yan Chen

Reviewer 2 Report
Please check the attachment.

English can be improve.
Author Response

(The authors gave the same response as above.)

Reviewer 3 Report
Dear Authors, I would like to suggest some suggestions for improving of the manuscript:
- Firstly, study area should be presented more detailed. Some description of selected counties in term of agriculture, tourism and socio-economic development need to be added for the better understanding of the case. What common or specific features have the selected counties? Also, the map with selected counties as well as some geographical and touristic information could help to the Readers.
- Secondly, some comments about integrated development of agriculture and tourism should be given. What does it mean "integrated development of agriculture and tourism" in Chinese case? Has this development some differences with similar territories in the World and/or in China?
- Thirdly, research Hypotheses need to be described more detailed in the conclusion section. Which of four were confirmed or refuted in the study? What kind of theoretical and practical recommendations could the Authors give based on these findings? The contributions of the Authors to the study of the integrated development of agriculture and tourism should be described more detailed.
Best regards, Reviewer
Author Response

(The authors gave the same response as above.)

Reviewer 4 Report
I have read the manuscript carefully and I believe that it deals with a topic relevant to China, which could have an impact beyond the scope of the presented specifics.
The research is detailed and thorough. The chosen methodology can be considered relevant to the object and subject of research. However, I have some remarks that in no way diminish the scientific and applied value of the research.
I have provided some specific comments in the manuscript file attached to the review.
Here I just want to add that I would recommend restructuring the text in the part of Sections 4 and 5. Section 4 is titled Data and Methodology, however there are some results presented; and Section 5 is devoted to a discussion of the empirics. In my opinion, it is indispensable to distinguish the methodology from the discussion, so that at the end of section 5, clear dependencies between the considered parameters can be drawn. As well as clearly showing the result of proving/rejecting the hypotheses. Currently, section 5 ends with a table and this creates a feeling of incompleteness.

In the text, there are sentences that should be edited because they do not sound precise in English. I would recommend that the authors pay attention to the use of present historical tense and avoid the pronoun "we". I would recommend that the text be reviewed once more for grammatical corrections.
I have noted some of the perceived weaknesses in the manuscript file.
Author Response

(The authors gave the same response as above.)

Round 2
Reviewer 1 Report
1. Please reconsider the research framework diagram. I believe that this is a part that needs to be revised.
2. Can you clarify if you personally drew Figures 2 and 3 in the article? If not, please provide proof of authorization.
3. Although the English in this research can be understood, the overall writing logic still follows the logic of Chinese-style English.
Although the English in this research can be understood, the overall writing logic still follows the logic of Chinese-style English.
Author Response

(The authors gave the same response as above.)
